# Fipronil in sub-lethal doses leads to immuno-toxicological effects in broiler birds

**Shafia Tehseen Gul**[1], **Muhammad Zergham Tahir**[1], **Latif Ahmad**[2], **Aisha Khatoon**[1],
**Muhammad Kashif Saleemi**[1], **Farid Shokry Ataya**[3], **Riaz Hussain**[4], **Bakhtawar Maqbool**[5],
**Dalia Fouad**[6], **Ahrar Khan**[1,7] *

1 Department of Pathology, Faculty of Veterinary Science, University of Agriculture, Faisalabad, Pakistan,
2 Pathology Department, Baqai Medical University (Veterinary Campus), Karachi, Pakistan, 3 Department of
Biochemistry, College of Science, King Saud University, Riyadh, Saudi Arabia, 4 Department of Pathology,
University College of Veterinary and Animal Sciences, The Islamia University of Bahawalpur, Bahawalpur,
Pakistan, 5 Institute of Microbiology, Faculty of Veterinary Science, University of Agriculture, Faisalabad,
Pakistan, 6 Department of Zoology, College of Science, King Saud University, Riyadh, Saudi Arabia,
7 Shandong Vocational Animal Science and Veterinary College, Weifang, China

* ahrar1122@yahoo.com

**Data Availability Statement:** The data analysed/generated during the current study are available from the website https://tjj.ln.gov.cn/tjj/tjxx/xxcx/tjnj/index.shtml or included in this published article.

## Abstract

Pesticides, including fipronil, are used mainly in agriculture; however, in veterinary and animal husbandry, their potential use is to control the pests responsible for vector-borne diseases. Their residues in agriculture products and direct use on farms are responsible for potentially harming livestock and poultry. So, this study was designed to evaluate the toxico-pathological effects of fipronil on the immune system of poultry birds. One hundred a-day-old chicks were purchased from a local hatchery, and standard housing conditions were provided from brooding till the end of the trail. The temperature at brooding was kept at 33˚C; later on, it was maintained at 26–28˚C, and the humidity was at 60–70%. Clean water and a basal diet were provided *ad libitum*. After three days of acclimatization, birds were divided into five experimental groups (A to E), each containing 20 birds. Group A was kept as a control group. Fipronil was administered orally through crop tubing @ 1.5, 2.5, 3.5, and 4.5 mg/kg to groups B-E, respectively. Birds were euthanized humanely on the 15th, 30th, and 45th days of the experiment. Immunological parameters were evaluated, i.e., antibody titers against NDV and SRBCs, phagocytic activity to clear carbon particles, avian incompetence to avian tuberculin, and histopathological alterations in the lymphoid organs. The SAS® University Edition software was used for data analysis. The results indicated decreased antibody titers against NDV in the treatment groups compared to the control. Similarly, antibody response to SRBCs, phagocytic activity in clearing the carbon particles, and sensitivity to avian tuberculin in the treatment groups were also decreased. Results also revealed that the bursa of Fabricius thymus and spleen were also affected due to the toxic effect of fipronil, even at sub-lethal doses.

## Introduction

Pesticides are used in advanced health practices, agriculture, and animal husbandry. Around the globe, the extensive use of these chemicals poses a serious threat to the environment and

**Funding:** This study was supported by King Saud University (RSPD2024R693 to FSA).

**Competing interests:** The authors have declared that no competing interests exist.

human and animal health [1–3]. Although due to such potentially harmful effects, pesticides and insecticides have been limited gradually in many countries like the European Union, USA, and China, but still in developing regions, they are being used extensively [4]. Fipronil is from the class of phenyl pyrazoles and was initially discovered in 1987 and was launched in 1993. The half-life of fipronil in granular form in aqueous form and soil is 125 and 438 hours, respectively. Being a soil or water contaminant, just like many other pesticides and insecticides, it has been reported to lead to oxidative stress and injuries in poultry, livestock, and humans [5, 6].

Fipronil is a wide-spectrum insecticide that hits γ-aminobutyric acid (GABA)-gated chloride channels and causes insect death. The vertebrates' organs affected by fipronil subjection include the thyroid, spleen, kidney, and liver [7, 8]. The toxicity of fipronil is highly selective, and it has a better affinity than mammalian receptors with arthropod GABA. The links to the GABA receptor interfere with chloride channels, leading to loss of neuronal signals, over-awakening, and death [9]. There are three major pathways by which fipronil is metabolized in the environment through photolysis on the upper surface of the soil, hydrolysis, or oxidation, and leads to the formation of fipronil sulfide, fipronil amide, and fipronil sulfone, respectively. Except for fipronil amide, the other two are more persistent and toxic in the environment [10–12].

Due to its extensive application of insecticides/pesticides, such as pyrethroids, organophosphates, and carbamates in agricultural systems to control pests lead residues in crops, soil, groundwater, rivers, etc. This residual effect plays a pivotal role in environmental pollution and its potentially harmful effects on non-target species, including bees, vertebrates, aquatic species, and poultry. Many non-target species are part of food chains that severely threaten human health [4, 13, 14]. Although fipronil is present in the environment at low concentrations, its chronic exposure is much more likely to occur than acute poisoning to non-target species that require a higher level to have an observable effect [6, 7, 15].

Many environmental samples contain the residues of fipronil or its metabolites, and they have been detected in soil, water, vegetables, and animal products due to the mixing of irrigation water with other sources of water [5, 11]. In animals, fipronil has been detected in birds' adipose tissue, brain, and liver after chronic exposure. It has been reported that birds exposed to a sublethal dose of fipronil result in a histopathological alteration in the spleen, liver, and kidney and a genotoxic effect. Fipronil-associated damage to DNA and the histopathological changes in tissues, mainly at high doses, have been published [16, 17]. In a few experimental studies where fipronil toxicities were induced, higher morbidity and mortality rates have been recorded, along with reduced body size, decreased hatching, etc. In addition, heart deformities such as edema and irregular heartbeat have been documented due to abnormal blood flow [6, 18, 19].

In fish, fipronil exposure generally leads to erythrocyte injury, immune system suppression, tissue physiology disruption, and death [20]. Fipronil has been declared harmful to many birds and most fish species. This insecticide has sub-lethal effects that result in concentrations far below those associated with deaths, varying from genotoxic and cytotoxic impact, compromising the immune system and reproduction rate. Using fipronil as a seed treatment on some crops poses risks for small birds, and the intake of only a few treated seeds could lead to mortality or fertility problems for sensitive bird species [21].

If vertebrates are exposed to fipronil, unwanted effects have been recorded, like intense vascular congestion in the liver during histopathological evaluation. Fipronil disrupts the immune system, leading to decreased cellular immunity and a reduction in the weight of lymphoid organs, spleen, and thymus in terms of organ-to-body weight ratio [22–24]. It has been reported that if mothers are exposed to fipronil, it leads to the marked induction of apoptosis

in the offspring's thyroid and thymus [25, 26]. In mammals, fipronil has been reported to result in a decline in immunoglobulin M (IgM) levels and different histopathological alterations [27].

Histopathological alteration has also been observed in the vertebrates' liver, gill, kidney, and brain. The liver displayed other damages, including deposition, pyknosis, steatosis, hypertrophy, sinusoidal dilation, and glycogen [4, 28]. The most critical damages found were renal tubular degeneration, sinusoidal dilation, systemic changes, and hemolysis in the kidneys. These effects were more apparent in the maximal dose, while the duration of exposure to this organ had less impact on fipronil toxicity [29].

Even though it is used extensively in agriculture, fipronil is also used in veterinary medicine as a broad-spectrum insecticide to kill ectoparasites, including lice, beetles, ticks, and cockroaches. However, it led to a global food safety incident in 2017. It was found in chicken eggs in more than 40 countries [13, 30]. Residues in milk and meat of bovines have also been detected. Considering all these harmful effects at toxic levels, the current study was planned to evaluate the potential harmful effects at sub-lethal dosage levels, being a food chain contaminant in broiler birds that are being used as a good and cost-effective source of animal protein in Pakistan.

## Materials and methods

### Ethical approval

This experimental study was conducted after the thorough approval from the Graduate Studies and Research Board, University of Agriculture, Faisalabad (UAF), Pakistan vide letter No. 10917–20, Dated 14-05-2020. All husbandry practices and euthanasia were performed, keeping in view all considerations of animal welfare.

### Chemicals

Fipronil ($C_{12}H_4Cl_2F_6N$) belongs to the phenylpyrazole chemical family and is a commercial product (REFREE® 0.3%G by Kanzo). It was purchased from a local market in Faisalabad, Pakistan, with a purity of 30% and stored at room temperature.

### Experimental birds and treatments

One-day-old broiler chicks (100) Ross 308 were purchased from the hatchery and maintained in a standard housing environment. The temperature was maintained at 37°C and gradually decreased to 25°C until the end of the experiment, with 60–70% humidity. At the start of the experiment, 22 hours of light were provided, and 20 hours until the end of the experiment. Clean water and a basal diet were provided (*ad libitum*). Birds were vaccinated for the diseases according to the prescribed schedule for broilers. All the birds were acclimatized for three days and then randomly divided into five experimental groups (20 birds/group).

LD50 of fipronil for avian species has been mentioned as 11.3 [31]. Group A served as control. Groups B, C, D, and E received fipronil @ 1.5, 2.5, 3.5, and 4.5 mg/kg body weight (BW) daily, and these doses were selected based on previous studies reported [32]. The fipronil was dissolved in corn oil and administered to each bird through a crop tube. Birds were monitored twice daily to observe behavioral changes, if any. To evaluate the humoral immune response, these birds were vaccinated against Newcastle disease (ND) at 5, 23, and 28 days of age (ND Lasota strain, VRI, Lahore, Pakistan). This trial duration was 45 days. During this trial, no mortality was recorded. Birds were shifted to the necropsy unit in the Department of Pathology to collect samples for the histopathology on the 15th, 30th, and 45th days of the

experiment. Every time, 6 birds were selected for necropsy and blood sample collections at each sampling phase. Blood without anticoagulant was collected from the wing vein in gel clot tubes for the serum samples, and then the serum was separated and stored until analysis. Euthanasia was performed as per the recommendation of the animal ethics committee of the University of Agriculture, Faisalabad, Pakistan. We used manual cervical dislocation which is the most common method for euthanizing broiler chickens. Later on, a necropsy was conducted as per the standard protocols for poultry birds.

## Parameters studied

### Immunological studies

**Antibody titers against Newcastle disease virus.** The procedure already described was followed to determine the antibody titers against the ND virus [33]. Chicks were vaccinated against ND at the age of 2, 23, and 28 days to assess the humoral responses of the treated birds. Serum was extracted from blood samples collected on the experimental trial's 14th, 21st, 28th, and 35th days. Serum samples were then subjected to the hemagglutination inhibition (HI) assay [34] to determine the degree of antibody formation against the ND in these fipronil birds.

**Antibody titer against sheep RBCs.** The trial evaluated humoral immune responses by estimating antibody titer against sheep red blood cells (SRBCs) [35]. Briefly, 1mL of a 3% SRBCs suspension was injected I/V into three chicks from each treated group (i.e., primary dosing) at the age of Day 7. A booster dose was injected 14 days later into these birds (i.e., at Day 21 of age). The wing vein was used to collect blood without anti-coagulant on 7, 14, 21, and 28 days post-primary doses for serum collection. Each serum sample was stored at -20°C until analysis. Serum samples were also inactivated for 30 minutes at 56°C and then analyzed for total anti-SRBCs antibodies following the method already described [35].

**Lymphoproliferative response to avian tuberculin.** The cell-mediated immune response was assessed through delayed-type hypersensitivity response to avian tuberculin [36] through lymphoproliferative response. For this test, at day 40 of age, three birds/group received an injection (0.2mL avian tuberculin; Veterinary Research Institute, Lahore, Pakistan) into the intra-digital space between the 3rd and 4th digit of their right foot. Normal saline was injected into the same space in the left foot as the "self"-control. Skin thickness between the digits was measured 24, 48, and 72 hours post-injection. Thickness was also measured before either injection to provide a baseline value. The cutaneous hypersensitivity response at each time point was calculated.

**Carbon clearance assay.** A non-specific immune response was accessed through a carbon clearance assay for phagocytic activity to clear the carbon particles [37]. For this test, on day 14 of age, three chicks/group received ink (Pelikan ink, 1 mL/kg) injection into the right-wing vein. An identical treatment was carried out on another subset set of three chicks/group at day 28 of age. On each day, blood samples were collected from the left-wing vein at 0 minutes, 3 minutes, and 15 minutes post-ink injection. One mL of blood from each sample was instantly mixed with 4 mL of 1% sodium citrate and centrifuged for 4 minutes. The relative amount of carbon particles remaining in the supernatant was then measured with a spectrophotometer (Spectro 20D Plus RS-232C) at 640nm [37].

**Gross and histopathology.** To evaluate toxicological effects on immune organs, including the bursa of Fabricius, spleen, and thymus, these organs' absolute and relative weights were recorded on the 15, 30, and 45th days of the experiment. For this purpose, birds from each group were euthanized humanely through cervical dislocation, and lymphoid organs were collected. Absolute organ weight was measured. Relative organ weight was calculated as % (organ

weight x 100 / live body weight). Tissue samples collected from the above-mentioned lymphoid organs were collected and preserved for histopathology in a neutral 10% buffered formalin solution [38], dehydrated, embedded, mounted on glass slides, and stained with hematoxylin and eosin (H&E) staining method as per following the procedure already described [39, 40].

**Statistical analysis.** Data thus collected during this experimental trial were subjected to ANOVA (analysis of variance) to determine the statistical difference between the means, and these means were compared using the Tukey post hoc test to determine the highest significant difference using SAS® University Edition online software SAS stat 15.1. All the results obtained during the trial have been expressed as means with their standard error. Regarding the homogeneity of ANOVA, Levene's test indicated a non-significant change at a significance level of 0.05, which is considered homogeneous [41].

## Results

### Immunological parameters

**Antibody titers against Newcastle disease virus.** The results recorded on the 14th and 21st day of the experiment indicated that birds of Group A (control) had significantly ($p < 0.05$) higher antibody titers against NDV antibody titers against NDV as compared to birds in Groups C (Fipronil @ 2.5 mg/kg BW), D (Fipronil @ 3.5 mg/kg BW), and E (Fipronil @ 4.5 mg/kg BW). In contrast, these groups were not significant among each other. Birds of Group B (Fipronil @ 1.5 mg/kg BW) had non-significant ($p > 0.05$) lower antibody titers against NDV as compared to control Group A (Table 1). Similar trends and results were observed on days 28th and 35th, as shown in the data presented in Table 1.

**Total antibody titers against SRBCs.** After assessing the total antibody titers against SRBCs, the results indicated that on the 7th day of primary injection, birds in control Group A (control) showed the highest antibody titers against SRBCs. Birds of Group E (Fipronil @ 4.5 mg/kg BW) had a significant ($p < 0.05$) decrease in antibody titers against SRBCs compared to birds in the control group. Birds in Groups B (Fipronil @ 1.5 mg/kg BW), C (Fipronil @ 2.5 mg/kg BW), and D (Fipronil @ 3.5 mg/kg BW) showed a non-significant decrease in antibody titers against SRBCs from the control group (Table 2). However, on the 7th day of booster injection, the control group (A) led the highest antibody titers against SRBCs. At the same time, birds of Groups B, C, D, and E showed a significant ($p < 0.05$) decrease in antibody titers against SRBCs compared to birds in the control group. However, these birds in groups (B-E) had non-significantly different antibody titers against SRBCs to each other (Table 2).

On the 14th day of primary and booster injection in birds of the control group (A) led to significantly higher antibody titers against SRBCs as compared to birds in Groups C (Fipronil @

**Table 1. Antibody titers of the broiler birds against Newcastle disease virus treated with different doses of fipronil.**

| Groups | Experimental Days | | | |
|---|---|---|---|---|
| | Day 14th | Day 21st | Day 28th | Day 35th |
| A (control) | 170.83 ± 43.60[a] | 213.33 ± 73.89[a] | 106.67 ± 16.91[a] | 85.33 ± 8.64[a] |
| B | 106.17 ± 34.24[ab] | 117.83 ± 21.92[ab] | 26.83 ± 9.28[b] | 53.65 ± 8.05[ab] |
| C | 42.67 ± 17.99[b] | 42.83 ± 17.72[ab] | 13.83 ± 4.61[b] | 5.33 ± 2.30[bc] |
| D | 37.33 ± 22.64[b] | 18.33 ± 12.40[b] | 14.67 ± 5.14[b] | 5.33 ± 2.29[bc] |
| E | 21.67 ± 9.16[b] | 10.83 ± 4.67[b] | 5.17 ± 2.32[b] | 3.33 ± 1.12[c] |

Values (Mean ± SD) bearing different letters in a column differ significantly (P≤0.05). Groups A, B, C, D, and E received fipronil @ 0 (control), 1.5, 2.5, 3.5, and 4.5 mg/kg body weight daily through crop tubing for 45 days.

**Table 2. Antibody titers against SRBCs of broiler birds treated with different doses of fipronil.**

| Parameter/ Groups | Day 7th (after primary injection) | Day 14th (after primary injection + booster dose) | Day 21st (after 7th day of booster dose) | Day 28th (after 14th day of booster dose) |
|---|---|---|---|---|
| Total Antibody Titers against SRBCs | | | | |
| A (control) | 7.33 ± 1.54[a] | 7.33 ± 0.54[a] | 8.33 ± 1.15[a] | 5.66 ± 1.15[a] |
| B | 6.33 ± 1.50[ab] | 5.66 ± 0.58[ab] | 5.66 ± 0.58[b] | 4.33 ± 0.57[ab] |
| C | 5.33 ± 1.47[ab] | 4.33 ± 1.54[bc] | 5.33 ± 0.64[b] | 3.00 ± 0.45[bc] |
| D | 4.66 ± 0.53[b] | 3.66 ± 0.55[bc] | 4.33 ± 0.58[b] | 2.33 ± 0.55[c] |
| E | 4.00 ± 0.54[b] | 2.66 ± 0.54[c] | 4.00 ± 0.57[b] | 1.66 ± 0.59[c] |
| IgG values against SRBCs | | | | |
| A (control) | 4.00 ± 1.00[a] | 3.66 ± 1.53[a] | 4.33 ± 1.50[a] | 2.66 ± 0.57[a] |
| B | 3.66 ± 1.14[ab] | 3.33 ± 0.57[ab] | 3.34 ± 0.57[ab] | 1.67 ± 0.56[ab] |
| C | 3.33 ± 2.08[ab] | 2.66 ± 0.57[ab] | 3.33 ± 0.57[ab] | 1.66 ± 0.56[ab] |
| D | 2.66 ± 0.57[ab] | 2.33 ± 0.56[ab] | 3.00 ± 1.06[ab] | 1.00 ± 0.57[b] |
| E | 2.31 ± 0.57[ab] | 1.33 ± 0.57[b] | 2.66 ± 0.58[ab] | 1.01 ± 0.58[b] |
| IgM values against SRBCs | | | | |
| A (control) | 3.66 ± 0.57[a] | 3.66 ± 1.52[a] | 4.00 ± 0.63[a] | 3.00 ± 1.00[a] |
| B | 2.66 ± 0.58[ab] | 2.33 ± 0.57[ab] | 2.33 ± 0.57[ab] | 2.67 ± 0.57[a] |
| C | 2.00 ± 1.00[ab] | 1.66 ± 1.13[ab] | 2.00 ± 0.58[ab] | 1.33 ± 0.57[ab] |
| D | 2.00 ± 0.56[ab] | 1.33 ± 0.58[ab] | 1.34 ± 0.57[ab] | 1.33 ± 0.55[ab] |
| E | 1.66 ± 0.58[b] | 1.33 ± 0.55[ab] | 1.33 ± 0.57[ab] | 0.66 ± 0.57[b] |

Values (Mean ± SD) bearing different letters in a column under specific parameters differ

significantly ($P \leq 0.05$). Groups A, B, C, D, and E received fipronil @ 0 (control), 1.5, 2.5, 3.5, and 4.5 mg/kg body weight daily through crop tubing for 45 days.

2.5 mg/kg BW), D (Fipronil @ 3.5 mg/kg BW) and E (Fipronil @ 4.5 mg/kg BW); however, these had non-significantly different antibody titers against SRBCs among each other. Group B (Fipronil @ 1.5 mg/kg BW) had lower non-significant ($p > 0.05$) antibody titers against SRBCs from the control group (Table 2).

**IgG response.** After the 7th day of primary injection, birds of the control group (A) showed the highest IgG response (4.00 ± 1.00); however, IgG in birds of Groups B, C, D, and E (Fipronil @ 1.5, 2.5, 3.5 and 4.5 mg/kg BW, respectively), showed a non-significant difference as compared to the control group (Table 2). After the 14th day of primary injection, birds in the control group showed the highest IgG response against SRBCs. Birds in Groups B, C, and D (Fipronil @ 1.5, 2.5, and 3.5 mg/kg BW, respectively) showed a non-significant decrease in IgG response as compared to control Group A, while Group E (1.33 ± 0.57) showed a significant ($p < 0.05$) reduction in IgG response as compared to birds in control and other groups (Table 2).

After the 7th day of booster injection, birds of the control group (4.33 ± 1.50) showed a higher IgG response; however, IgG in birds of Groups B, C, D, and E (Fipronil @ 1.5, 2.5, 3.5, and 4.5 mg/kg BW, respectively) showed a non-significant difference as compared to the control group (Table 2). On the 14th day of booster injection, the highest IgG response was observed in birds of the control group, whereas Groups D and E (Fipronil @ 3.5, and 4.5 mg/kg BW, respectively) had a significant ($p < 0.05$) decrease in IgG response compared to the control group. However, Groups B and C (Fipronil @ 1.5, and 2.5 mg/kg BW, respectively) birds were non-significantly different from the control group (Table 2).

**IgM response.** As mentioned above, the IgM response was similar to that of IgG in fipronil-treated groups. Briefly, on day 7th, birds in Group E (1.66 ± 0.58) showed significantly ($p < 0.05$) the lowest IgM response compared to control birds in Group A (3.66 ± 0.57). Birds

in Groups B, C, and D (Fipronil @ 1.5, 2.5, and 3.5 mg/kg BW) showed a non-significant decrease in IgM response compared to control group A. A similar trend was observed after the 14th day of primary injection data (Table 2).

After day 7th of booster injection, birds in Groups B, C, D, and E (Fipronil @ 1.5, 2.5, 3.5, and 4.5 mg/kg BW, respectively) showed a non-significant ($p > 0.05$) decrease in IgM response against SRBCs compared to the control group. However, on the day 14th of booster injection, birds in Group E ($0.66 \pm 0.57$) showed significantly ($p < 0.05$) lower IgM response as compared to birds of control Group A. Birds in Groups B, C, and D (Fipronil @ 1.5, 2.5, and 3.5 mg/kg BW, respectively) showed a non-significant decrease in IgM response as compared to birds in the control group (Table 2).

**Incompetence to avian tuberculin.** Avian tuberculin was injected into the birds, and skin thickness (mm) was measured. The result indicated that after 24 hours, birds of Groups C, D, and E (Fipronil @ 2.5, 3.5, and 4.5 mg/kg BW, respectively) had significantly ($p < 0.05$) lower values as compared to birds in the control group ($1.25 \pm 0.12$). Still, the response to avian tuberculin was non-significantly ($p > 0.05$) different from birds of fipronil-treated birds. The skin thickness of birds in Group B ($1.04 \pm 0.11$) showed a non-significant decrease compared to birds in the control group. However, it has significantly ($p < 0.05$) higher skin thickness than groups C ($0.94 \pm 0.09$), D ($0.74 \pm 0.10$) and E ($0.60 \pm 0.09$) (Fig 1).

After 48 hours of avian tuberculin injection, birds of groups D and E (Fipronil @ 3.5, and 4.5 mg/kg BW, respectively) had significantly ($p < 0.05$) lower thickness as compared to the birds in the control group ($0.66 \pm 0.16$). Groups B and C (Fipronil @ 1.5 and 2.5 mg/kg BW, respectively) birds showed a non-significant decrease in skin thickness compared to the control group. After 72 hours, significantly ($p < 0.05$), the lowest skin thickness values were recorded in group E ($0.19 \pm 0.04$) as compared to all treatment groups. The birds in fipronil-treated groups had significantly ($p < 0.05$) lower skin thickness as compared to birds in the control group but were non-significantly different from each other (Fig 1).

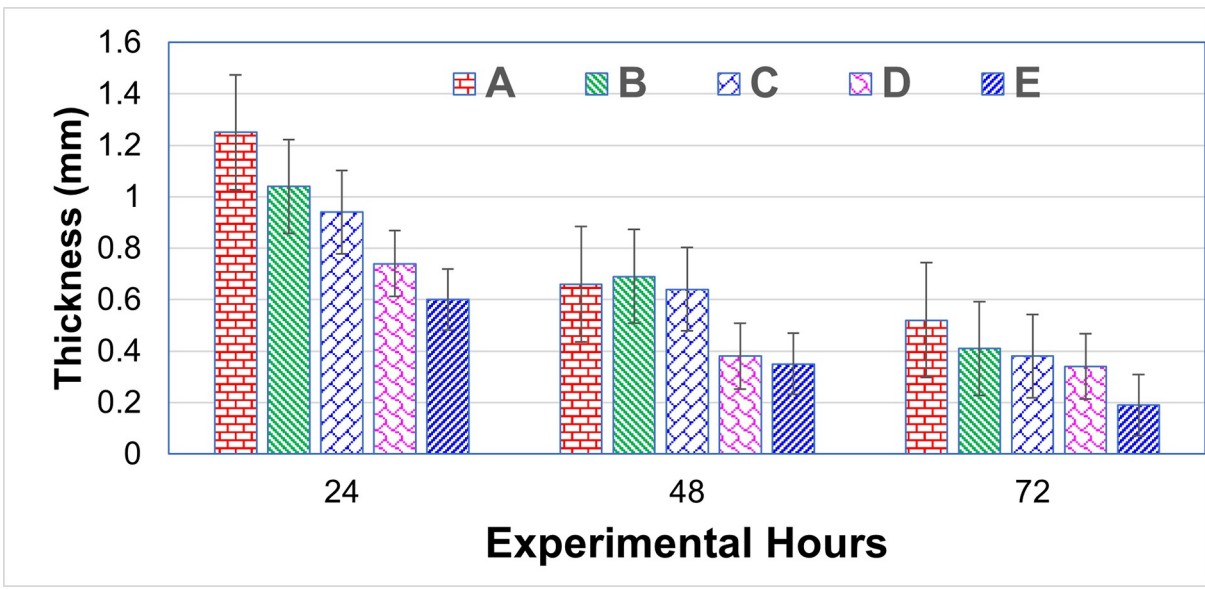

**Fig 1. Lymphoproliferative response against avian tuberculin in different treatment groups against fipronil toxicity.** Groups A, B, C, D, and E received fipronil @ 0 (control), 1.5, 2.5, 3.5, and 4.5 mg/kg body weight daily through crop tubing for 45 days. The cell-mediated immune response was assessed through a delayed-type hypersensitivity response to avian tuberculin following the method already described [34].

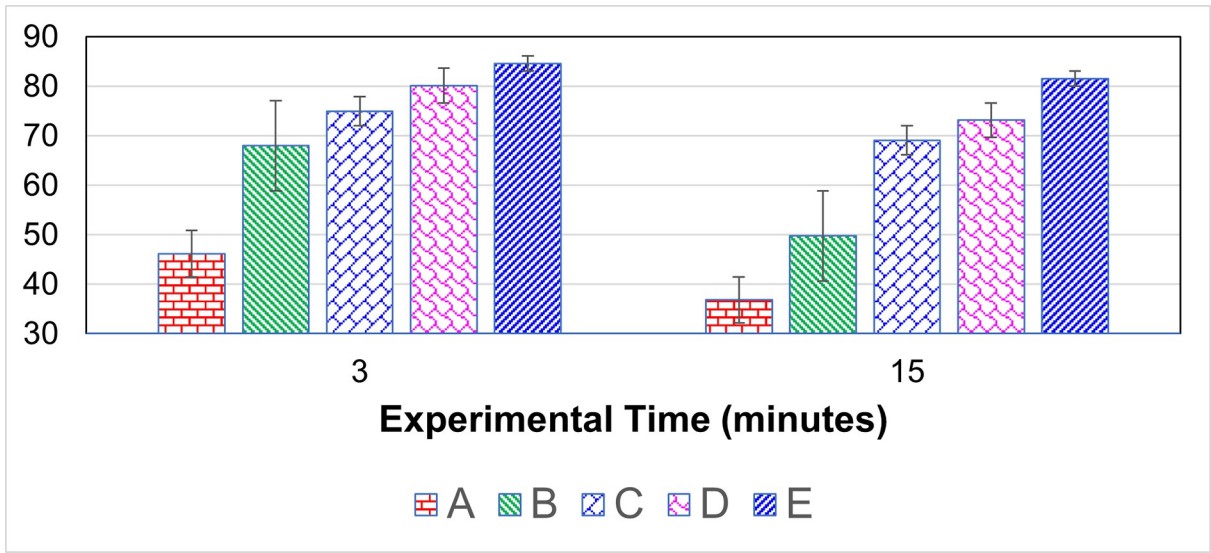

**Fig 2. Phagocytic activity in clearing carbon particles treated with fipronil.** Groups A, B, C, D, and E received fipronil @ 0 (control), 1.5, 2.5, 3.5, and 4.5 mg/kg body weight daily through crop tubing for 45 days.

**Phagocytic activity through carbon clearance assay (CCA).** Pelikan ink was injected in the wing veins of the birds to determine the phagocytic activity in clearing the carbon particles. Results obtained after 3 minutes indicated that birds in Group E (84.61 ± 1.82) had significantly ($p < 0.05$) higher values as compared to the birds in control Group A (46.16 ± 5.63) and other fipronil-treated groups (Fig 2). Follow-up reading was recorded after 15 minutes. That indicated the highest carbon particles in Group E (81.57 ± 4.15) followed by Groups D and C (Fipronil @ 3.5 and 2.5 mg/kg BW, respectively); however, statistically, these values were non-significant to each other. However, birds in Group A (36.85 ± 2.52) showed significantly ($p < 0.05$) different Indian ink responses than all the fipronil-treated Groups, but a non-significant difference was recorded to that of Group B (Fig 2).

## Immune organ weight

**The absolute and relative weight of the spleen.** The results revealed that the absolute weight of the spleen was inversely proportional to the dose of the pesticide; as the dose of the pesticide increased, the weight of the spleen decreased. On day 15th (0.53 ± 0.07), 30th (1.88 ± 0.05), and 45th (4.08 ± 0.06) weight of the spleen of control Group A had significantly ($p < 0.05$) higher weights as compared to other fipronil treated groups (Table 3). On the 15th, 30th, and 45th day of the experiment indicated that the control group (group A) had non-significantly ($p > 0.05$) higher relative weight from all fipronil-treated groups, i.e., Groups B, C, D, and E (Fipronil @ 1.5, 2.5 and 3.5 mg/kg BW, respectively) (Table 3).

**The absolute and relative weight of the bursa of Fabricius.** The absolute weight of the bursa of Fabricius is also inversely proportional to the dose of fipronil, just like that of the spleen. On days 15th, 30th, and 45th, the bursa of Fabricius of control Group A had significantly ($p < 0.05$) higher weights as compared to other fipronil-treated groups (Table 3). The relative organ weight results recorded on day 15th indicated that control Group A (0.18 ± 0.02) had significantly higher relative weight than Groups D and E (Fipronil @ 3.5 and 4.5 mg/kg BW, respectively) and non-significantly higher than Groups B and C (Fipronil @ 1.5 and 2.5 mg/kg BW, respectively). On day 30th, control Group A (Fipronil @ 0 mg/kg BW) had significantly ($p < 0.05$) higher relative weight than Group E, while significantly higher than Groups B, C,

**Table 3. Immune organs' absolute and relative weight in birds different experimental days organs treated with different doses of fipronil.**

| Organs/ Groups | 15th day | | 30th day | | 45th day | |
|---|---|---|---|---|---|---|
| | Absolute weight | Relative weight | Absolute weight | Relative weight | Absolute weight | Relative weight |
| **Spleen** | | | | | | |
| A(Control) | 0.53 ± 0.07[a] | 0.11 ± 0.02[a] | 1.88 ± 0.05[a] | 0.12 ± 0.03[a] | 4.08 ± 0.06[a] | 0.15 ± 0.04[a] |
| B | 0.44 ± 0.14[ab] | 0.10 ± 0.03[ab] | 1.19 ± 0.22[b] | 0.08 ± 0.02[ab] | 3.32 ± 0.54[ab] | 0.14 ± 0.03[ab] |
| C | 0.31 ± 0.01[b] | 0.07 ± 0.01[ab] | 1.03 ± 0.06[b] | 0.08 ± 0.01[b] | 2.78 ± 0.47[bc] | 0.12 ± 0.02[bc] |
| D | 0.26 ± 0.08[b] | 0.06 ± 0.02[ab] | 1.04 ± 0.28[b] | 0.07 ± 0.01[b] | 2.23 ± 0.18[c] | 0.11 ± 0.01[bc] |
| E | 0.23 ± 0.06[b] | 0.05 ± 0.01[ab] | 0.86 ± 0.12[b] | 0.06 ± 0.05[b] | 1.97 ± 0.26[c] | 0.10 ± 0.01[c] |
| **Bursa of Fabricius** | | | | | | |
| A(Control) | 0.94 ± 0.04[a] | 0.18 ± 0.02[a] | 0.88 ± 0.06[a] | 0.05 ± 0.03[a] | 0.99 ± 0.05[a] | 0.04 ± 0.01[a] |
| B | 0.74 ± 0.06[b] | 0.16 ± 0.04[ab] | 0.65 ± 0.11[ab] | 0.05 ± 0.01[ab] | 0.80 ± 0.07[ab] | 0.03 ± 0.01[ab] |
| C | 0.66 ± 0.05[bc] | 0.15 ± 0.05[ab] | 0.61 ± 0.10[b] | 0.04 ± 0.04[ab] | 0.60 ± 0.05[bc] | 0.03 ± 0.03[ab] |
| D | 0.55 ± 0.02[cd] | 0.13 ± 0.02[bc] | 0.56 ± 0.07[bc] | 0.03 ± 0.08[ab] | 0.45 ± 0.03[bc] | 0.02 ± 0.02[ab] |
| E | 0.45 ± 0.09[d] | 0.11 ± 0.02[c] | 0.37 ± 0.06[c] | 0.02 ± 0.03[b] | 0.38 ± 0.02[c] | 0.01 ± 0.01[b] |
| **Thymus** | | | | | | |
| A(Control) | 1.52 ± 0.12[a] | 0.30 ± 0.04[a] | 5.14 ± 0.15[a] | 0.32 ± 0.01[a] | 11.87 ± 0.70[a] | 0.45 ± 0.03[a] |
| B | 1.29 ± 0.27[b] | 0.29 ± 0.01[ab] | 4.16 ± 0.39[ab] | 0.30 ± 0.02[ab] | 9.03 ± 0.33[b] | 0.39 ± 0.01[ab] |
| C | 1.12 ± 0.17[bc] | 0.26 ± 0.02[ab] | 3.43 ± 0.50[ab] | 0.24 ± 0.03[ab] | 7.85 ± 0.64[bc] | 0.36 ± 0.02[bc] |
| D | 1.02 ± 0.74[cd] | 0.23 ± 0.06[ab] | 3.45 ± 0.63[ab] | 0.23 ± 0.01[ab] | 7.14 ± 0.97[cd] | 0.35 ± 0.04[bc] |
| E | 0.72 ± 0.41[d] | 0.17 ± 0.09[ab] | 2.66 ± 0.48[b] | 0.19 ± 0.02[ab] | 5.58 ± 0.69[d] | 0.29 ± 0.03[c] |

Values (Mean ± SD) bearing different letters in a column under specific parameters differ

significantly (P≤0.05). Groups A, B, C, D, and E received fipronil @ 0 (control), 1.5, 2.5, 3.5, and 4.5 mg/kg body weight daily through crop tubing for 45 days.

and D (Fipronil @ 1.5, 2.5 and 3.5 mg/kg BW, respectively). On day 45th, control Group A had significantly higher relative weight than Group E, while non-significantly different from Groups B, C, and D (Table 3).

**The absolute and relative weight of the thymus.** The results obtained for the thymus were similar to those mentioned above in two lymphoid organs. Briefly, on days 15th, 30th, and 45th, absolute weights of the thymus of control Group A were significantly ($p < 0.05$) higher weights as compared to others (Table 3). The results on the relative weight of the thymus revealed that on days 15th and 30th, control Group A had non-significantly higher relative weight than Groups B, C, D, and E (Fipronil @ 1.5, 2.5, 3.5, and 4.5 mg/kg BW, respectively). On day 45th, control Group A had significantly ($p < 0.05$) higher relative weight than Groups C, D, and E. In comparison, Group B had a non-significantly lower relative weight as compared to control Group A (Table 3).

## Histopathology of immune organs

**Spleen.** Group A (control) showed standard structure; no change was observed on the 15th, 30th, and 45th days. White pulp, red pulp, central arterioles, lymphoid follicle, and capsule showed no changes. Group B showed little disruption in central arterioles. However, the histological structure remains almost similar to the control group (A). Still, severe congestion was observed in Groups C and D (Fipronil @ 2.5 and 3.5 mg/kg BW, respectively). Group E, treated with a high dose of fipronil (4.5 mg/kg BW), showed lymphocyte depletion in the lymphoid follicles. The severity of the lesion increased in a dose-dependent manner and in the number of exposure days. Marked changes were recorded in the spleen of group E on the 30th and 45th day of the experiment (Fig 3).

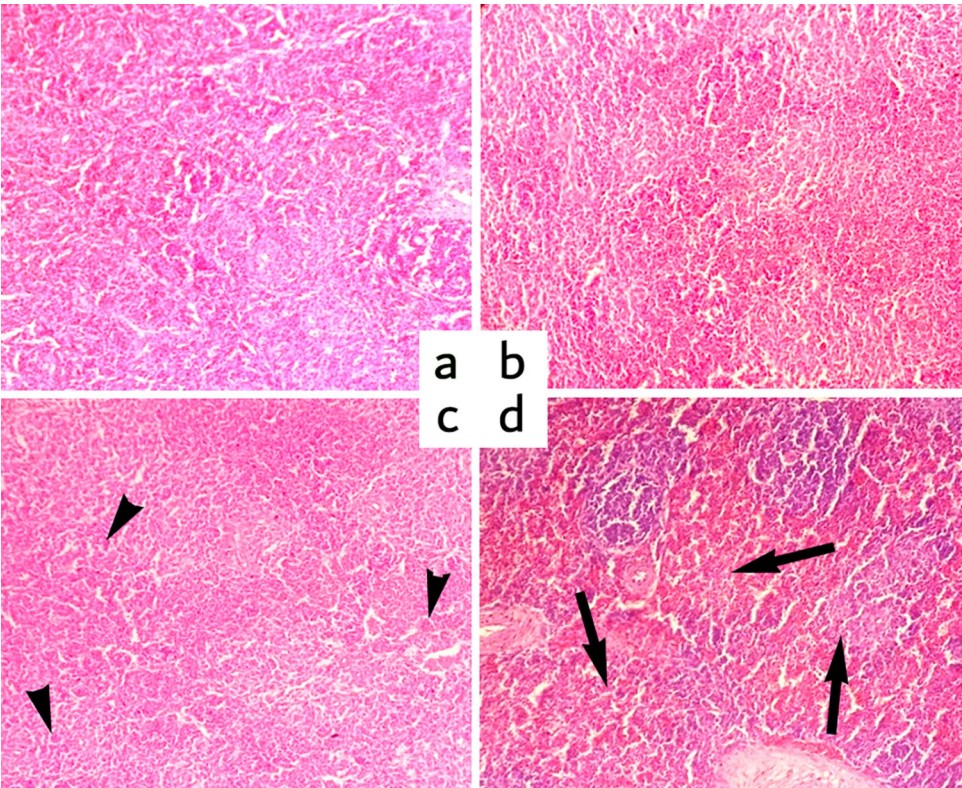

**Fig 3.** a) Photomicrograph of the spleen (Group A) at 15th day showing normal white and red pulp, b) Group E (Fipronil @ 4.5 mg/kg BW) at 15th day showing disruption in white and red pulp, c) Group E at 30th day showing severe congestion (arrowheads), and d) Group E at 45th depletion of lymphocytes (arrows). H & E Staining. Magnification 200X.

**Bursa of Fabricius.** Group A (Control) showed normal structure; no change was observed on experimental days. The lymphoid follicle showed the normal cortex, medulla, and cortico-medullary border. Group B (Fipronil 1.5 mg/kg BW), showed mild congestion and disruption of the cortex, medulla, and cortico-medullary border. However, severe congestion and disruption were observed in groups C and D (Fipronil @ 2.5 and 3.5 mg/kg BW, respectively). In the high dose fipronil (4.5 mg/kg BW) treated (group E), moderate infiltration of inflammatory cells was observed on day 15th. Laterally, vacuolar degeneration was also evident in the tissue collected on 30 and 45th day of the experiment (Fig 4). The severity of the lesions increased in a dose-dependent manner and duration of exposure. Marked histopathological changes were recorded in the bursa of the group E on the 30th and 45th day of the experiment (Fig 4).

**Thymus.** Group A (control) showed a normal cell structure. No change was observed on days 15th, 30th, and 45th. Clearly, demarcated Hassle's capsules were observed. Group B (Fipronil @ 1.5 mg/kg BW), showed mild changes and disruption, almost similar to A group (control), but groups C and D (Fipronil @ 2.5 and 3.5 mg/kg BW, respectively), showed severe congestion. Group E showed degenerative changes in the medullary region. In high-dose group E (fipronil: 4.5 mg/kg BW), marked histopathological changes in the thymus were recorded. On the 15th day, mild depletion of lymphocytic parenchyma was recorded, and on the 30th day, it was accompanied by vacuolar degeneration. Inflammatory and histopathological changes were also evident on the 45th day due to chronic exposure to fipronil.

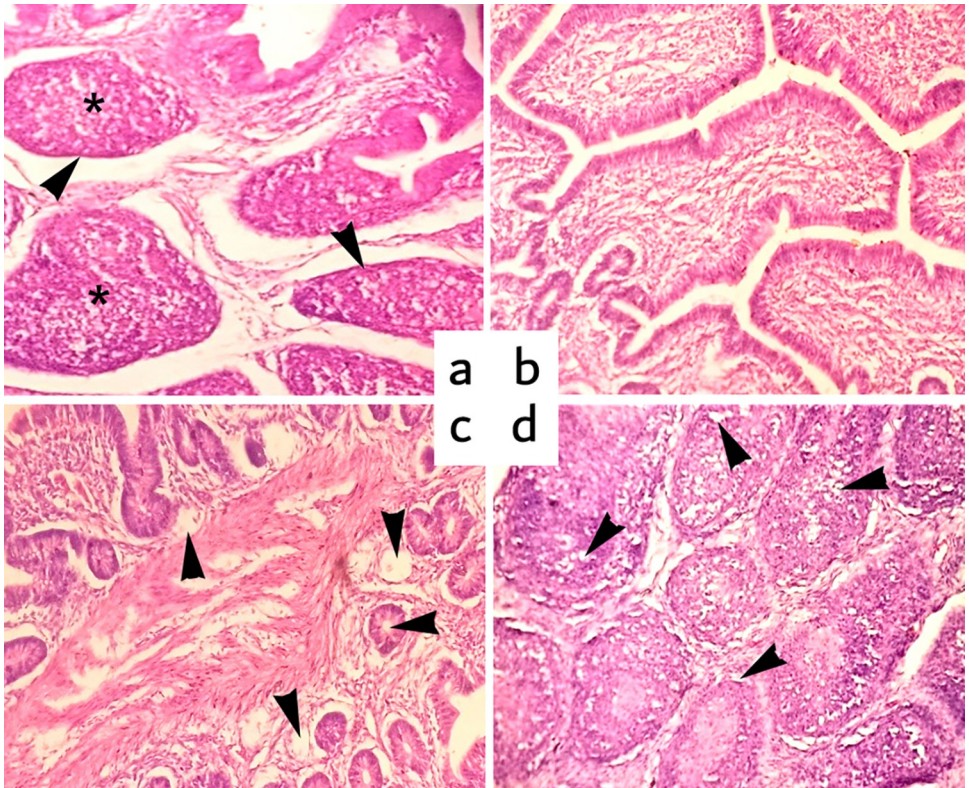

**Fig 4.** a) Photomicrograph of the bursa of Fabricius (Group A) at the 15th day showing normal cortex (arrowhead), medulla (asterisk), and cortico-medullary border, b) Group E (Fipronil @ 4.5 mg/kg BW) at 15th-day showing moderate infiltration of inflammatory cells, c) at Group E at 30th-day showing vacuolar degeneration (arrowheads), and d) Group E at 45th vacuolar degeneration (arrowheads) and infiltration of inflammatory cells. H & E Stain. Magnification 200X.

## Discussion

Poultry is a significant and vibrant agricultural sector in Pakistan. It contributes significantly to the national GDP (gross domestic product). The poultry sector gained commercial significance in Pakistan in the 1960s; since then, it has gained a significant state. This is a major and cheapest source of protein and contributes to approximately 30 percent of the country's overall meat production [42–45]. The main grains used in poultry feed are maize, wheat, rice, etc. Being a vibrant industry, the agriculture sector is continuously increasing the production of these crops. To ensure better crop yield, agrochemical companies are producing new forms of chemicals, but the danger to the consumer must be noticed. Such chemicals leave their crop residues behind and eventually enter the end-user [46, 47]. Despite high environmental effects, fipronil is still used in agriculture and veterinary setups [11, 48]. Due to its mechanism of action through receptor targeting, particularly for γ-aminobutyric acid, it leads to neurotoxicity, cytotoxicity, nephrotoxicity, and reproductive toxicity in vertebrates [49, 50].

In the current study, the effect of sub-lethal doses of fipronil was observed on the immunity and immune organs of the broiler birds. Significant results recorded during the trial have been discussed here. The antibody titers against NDV in fipronil-treated groups were significantly lower than those of the control group on the 14th, 21st, 28th, and 35th days of the trial. The lowest titers were recorded in group E (Fipronil @ 4.5 mg/kg BW). These results showed that even sub-lethal dosages of fipronil in birds depress immune systems, leading to immunosuppression and immunotoxicity [1, 20, 23].

Previously, humoral antibody production has been evaluated in different species through intravenous administration of SRBCs in chicken [33–35] against different pesticides/insecticides, and profound effects have been reported to be dose-dependent. In the current study, the effects of fipronil exposure on antibody titer against the SRBCs were also dose-dependent. The lowest antibody titers were observed in high-dose treated group E (Fipronil @ 4.5 mg/kg BW) on day 7[th] and so on. These results were consistent with earlier reports [51, 52]. The antibody titer to SRBCs was decreased due to inhibition of antigen by the reticuloendothelial system [53, 54] and to inhibition of immunoglobulin synthesis and reduction of T-lymphocyte [52].

Skin thickness was the highest lymphoproliferative response in the A group at 24, 48, and 72 hours. At 24 and 48 hours, significantly higher than Group E (Fipronil @ 4.5 mg/kg BW). Lymphoproliferative response against avian tuberculin was decreased because the T-cell response was lower, as previously reported [33, 54] against different xenobiotics, including chlorpyriphos, arsenic, and thiamethoxam (TMX), respectively. Intra-dermal injection of avian tuberculin stimulates the T and B cells. It increases the production of cytokines /chemokines and the recruitment of immune cells (macrophages, neutrophils, etc.), so skin thickness increases at the injection site [55].

In terms of delayed-type hypersensitivity immune response, it was observed that birds from control Group A had increased phagocytic activity in clearing the carbon particle, and birds from Group D and E (Fipronil @ 3.5 and 4.5 mg/kg BW, respectively) had decreased phagocytic activity in clearing the carbon particle. The phagocytic activity of carbon particles after 3 and 15 minutes was the highest in Group A (Fipronil @ 0 mg/kg BW), but the values were lowest, indicating delayed carbon clearance in all other treatment groups (B-E; Fipronil @ 1.5, 2.5, 3.5 and 4.5 mg/kg BW, respectively). The values increased, but the phagocytic response decreased when the fipronil dose rate was increased in treated groups (B-E). A similar result was examined after 15 min and 30 min, which are identical [52, 54]. The inflammatory response was decreased because the cells that reached the target site were immobilized. All the immune parameters observed during this trial have indicated that fipronil leads to immunotoxicity even at sub-lethal doses. It has been documented that fipronil induces immunotoxicity through free radical production in various tissues and mitochondrial injury that leads to cell death [56, 57]. The immune system plays a crucial role in maintaining the health system of humans and animals. Such xenobiotics, including fipronil and other pesticides, not only lead to immunosuppression but pose serious threats, including genetic abnormalities, hormonal disruption, cancer, etc. [28, 58].

The deleterious effects of fipronil have been reported on various organs. The result of the study is discussed below, as the immune organs were part of it. The results revealed that the weight of the spleen was inversely proportional to the dose of the pesticide; as the dose of the pesticide increased, the weight of the spleen decreased. The lowest absolute weight of the spleen was recorded in group E, and it was significantly lower on the experiment's 15[th], 30[th], and 45[th] days. However, the relative weight of the spleen on the 15[th] day of the experiment indicated that the control group had a non-significantly higher relative weight than all fipronil-treated groups, and the same trend was observed on the 30 and 45[th] days of the experiment. The other two organs, the thymus and bursa of the Fabricius, showed the same trend as that of the spleen in this trial. The results were per those previously reported by Bano and Mohanty [24], who noted that fipronil toxicity results in a decrease in functional parenchyma of the immune organs, leading to lower immune cell production, so the size of the immune organs is regressed as compared to the normal.

Histopathology of these organs was performed during this trial, and significant findings were recorded. Group A (control) showed the normal structure of the spleen; no change was observed on days 15[th], 30[th], and 45[th]. White pulp, red pulp central arterioles, and capsule

showed no changes. There is also a lymphoid follicle composed of B-lymphocytes. In high-dose treated group E, depletion of lymphocytes and infiltration of inflammatory cells were evident. The loss of lymphoid cells due to cell-mediated immune response suppression and decreased immune cell type also proposed that immunosuppression existed due to functional fault in immunocompetent cells [23].

Group A (Control) showed the normal structure of the bursa of Fabricius; no change was observed on experimental days 15th, 30th, and 45th. The lymphoid follicle showed the normal cortex (C), medulla (M), and cortico-medullary border. Group B showed mild congestion and disruption in the cortex (C), medulla (M), and cortico-medullary border, but severe congestion and disruption were observed in the C and D groups (Fipronil @ 2.5 and 3.5 mg/kg BW, respectively). In the last high-treated group, E showed severe congestion, infiltration of inflammatory cells, and increased follicular spacing. Edema, lymphocytic depletion in the medulla and cortex, and mild interfollicular fibrosis in the bursa of Fabricius due to pesticide toxicity have already been reported [51, 59].

Group A (Control) showed normal cell structures in the thymus. No changes were observed on days 15th, 30th, and 45th. Demarcated Hassall's capsules were present. Group B (Fipronil @ 1.5 mg/kg BW) showed mild changes and disruptions compared to group A (Control), but severe congestion was observed in the C and D groups (Fipronil @ 2.5 and 3.5 mg/kg BW, respectively). Group E (Fipronil @ 4.5 mg/kg BW) showed degenerative changes in the medullary region and depletion of lymphoid cells, and this difference can easily be identified in control group A.

Lymphocytes are related to adaptive immune responses, as these are involved in producing immunoglobulins and recruiting other cell types, thus modulating the immune response to specifically react to a given pathogenic challenge [60, 61]. Diminishing numbers of lymphocytes, therefore, directly impact the adaptive responses of animals to diseases. As such, the lymphopenia detected in *Odontophrynus carvalhoi* tadpoles exposed to chlorpyrifos reveals immunosuppression provoked by that xenobiotic, with consequent diminishing resistance to pathogens that could compromise their survival [61]. Similarly, most of the insecticides lead to leukopenia and lymphopenia, and sometimes, a decrease in the oxidative metabolic activity of phagocytes, which indicates immunosuppression [62–65]. Lymphopenia could be caused by insecticide destroying this cell series, resulting in lymphocyte loss [66, 67]. Cytotoxic effects of pesticides/insecticides on T-lymphocytes lead to the depletion of these lymphoid cells, which ultimately results in immunosuppression [68–71].

## Conclusion

It has been concluded that the fipronil toxicity at the sub-lethal level is also significant, leading to immunosuppression in non-target species when they are exposed to residues in the food chains. In poultry birds, these sub-lethal doses have immune-toxicological effects on humoral, cell-mediated, and non-specific immune responses. The toxic effects on the major immune organs in broilers in terms of decreased organ weight have also been recorded as playing a crucial role in immunosuppression.

## Supporting information

**S1 File. Supporting-data-files.**
(ZIP)

## Author Contributions

**Conceptualization:** Shafia Tehseen Gul, Muhammad Kashif Saleemi, Riaz Hussain.

**Data curation:** Shafia Tehseen Gul, Muhammad Zergham Tahir, Latif Ahmad, Aisha Khatoon, Riaz Hussain.

**Formal analysis:** Muhammad Zergham Tahir, Latif Ahmad, Aisha Khatoon, Muhammad Kashif Saleemi, Farid Shokry Ataya, Riaz Hussain, Bakhtawar Maqbool, Dalia Fouad.

**Investigation:** Muhammad Zergham Tahir, Aisha Khatoon, Muhammad Kashif Saleemi, Farid Shokry Ataya, Riaz Hussain, Dalia Fouad.

**Methodology:** Shafia Tehseen Gul, Muhammad Zergham Tahir, Aisha Khatoon, Bakhtawar Maqbool.

**Resources:** Ahrar Khan.

**Software:** Muhammad Kashif Saleemi, Bakhtawar Maqbool, Dalia Fouad, Ahrar Khan.

**Supervision:** Shafia Tehseen Gul, Muhammad Kashif Saleemi.

**Validation:** Latif Ahmad, Muhammad Kashif Saleemi, Farid Shokry Ataya, Riaz Hussain, Bakhtawar Maqbool, Dalia Fouad, Ahrar Khan.

**Visualization:** Muhammad Zergham Tahir, Dalia Fouad.

**Writing – original draft:** Shafia Tehseen Gul, Muhammad Zergham Tahir, Riaz Hussain.

**Writing – review & editing:** Latif Ahmad, Aisha Khatoon, Farid Shokry Ataya, Riaz Hussain, Dalia Fouad, Ahrar Khan.

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
