## [Decision Letter · Decision Letter 0]

5 Nov 2024

PONE-D-24-46735FIPRONIL IN SUB-LETHAL DOSES LEADS TO IMMUNO-TOXICOLOGICAL EFFECTS IN BROILER BIRDSPLOS ONE

Dear Dr. Khan,

Thank you for submitting your manuscript to PLOS ONE. After careful consideration, we feel that it has merit but does not fully meet PLOS ONE’s publication criteria as it currently stands. Therefore, we invite you to submit a revised version of the manuscript that addresses the points raised during the review process.

We look forward to receiving your revised manuscript.

Kind regards,

Adeel Sattar, Ph.D

Academic Editor

PLOS ONE

Journal Requirements: When submitting your revision, we need you to address these additional requirements. 1. Please ensure that your manuscript meets PLOS ONE's style requirements, including those for file naming. The PLOS ONE style templates can be found at https://journals.plos.org/plosone/s/file?id=wjVg/PLOSOne_formatting_sample_main_body.pdf and https://journals.plos.org/plosone/s/file?id=ba62/PLOSOne_formatting_sample_title_authors_affiliations.pdf 2. We note that your Data Availability Statement is currently as follows: If the data are all contained within the manuscript and/or Supporting Information files, enter the following: All relevant data are within the manuscript and its Supporting Information files. Please confirm at this time whether or not your submission contains all raw data required to replicate the results of your study. Authors must share the “minimal data set” for their submission. PLOS defines the minimal data set to consist of the data required to replicate all study findings reported in the article, as well as related metadata and methods (https://journals.plos.org/plosone/s/data-availability#loc-minimal-data-set-definition). For example, authors should submit the following data: - The values behind the means, standard deviations and other measures reported;- The values used to build graphs;- The points extracted from images for analysis. Authors do not need to submit their entire data set if only a portion of the data was used in the reported study. If your submission does not contain these data, please either upload them as Supporting Information files or deposit them to a stable, public repository and provide us with the relevant URLs, DOIs, or accession numbers. For a list of recommended repositories, please see https://journals.plos.org/plosone/s/recommended-repositories. If there are ethical or legal restrictions on sharing a de-identified data set, please explain them in detail (e.g., data contain potentially sensitive information, data are owned by a third-party organization, etc.) and who has imposed them (e.g., an ethics committee). Please also provide contact information for a data access committee, ethics committee, or other institutional body to which data requests may be sent. If data are owned by a third party, please indicate how others may request data access. 3. When completing the data availability statement of the submission form, you indicated that you will make your data available on acceptance. We strongly recommend all authors decide on a data sharing plan before acceptance, as the process can be lengthy and hold up publication timelines. Please note that, though access restrictions are acceptable now, your entire data will need to be made freely accessible if your manuscript is accepted for publication. This policy applies to all data except where public deposition would breach compliance with the protocol approved by your research ethics board. If you are unable to adhere to our open data policy, please kindly revise your statement to explain your reasoning and we will seek the editor's input on an exemption. Please be assured that, once you have provided your new statement, the assessment of your exemption will not hold up the peer review process. 4. Please include your full ethics statement in the ‘Methods’ section of your manuscript file. In your statement, please include the full name of the IRB or ethics committee who approved or waived your study, as well as whether or not you obtained informed written or verbal consent. If consent was waived for your study, please include this information in your statement as well. 5. Please review your reference list to ensure that it is complete and correct. If you have cited papers that have been retracted, please include the rationale for doing so in the manuscript text, or remove these references and replace them with relevant current references. Any changes to the reference list should be mentioned in the rebuttal letter that accompanies your revised manuscript. If you need to cite a retracted article, indicate the article’s retracted status in the References list and also include a citation and full reference for the retraction notice.

**Additional Editor Comments:**

Author are requested to address reviewers questions please

Reviewers' comments:

Reviewer's Responses to Questions

**Comments to the Author**

1. Is the manuscript technically sound, and do the data support the conclusions?

Reviewer #1: Yes

Reviewer #2: Yes

Reviewer #3: Yes

2. Has the statistical analysis been performed appropriately and rigorously? 

Reviewer #1: No

Reviewer #2: Yes

Reviewer #3: Yes

3. Have the authors made all data underlying the findings in their manuscript fully available?

Reviewer #1: Yes

Reviewer #2: Yes

Reviewer #3: Yes

4. Is the manuscript presented in an intelligible fashion and written in standard English?

Reviewer #1: Yes

Reviewer #2: Yes

Reviewer #3: Yes

5. Review Comments to the Author

Reviewer #1: major concern is how the data was analyzed for a dose-response design. Also, I recommend that the authors revise the Materials and Methods section to include comprehensive details on bird husbandry practices, sources of materials (e.g., fipronil, immune modulators), and specifics on treatment replications. Providing this information will ensure that future researchers can accurately replicate the study. Key aspects such as housing conditions, diet, dosing protocols, intervals between treatments, and any other procedural details should be clearly described to enhance the study’s transparency and replicability. Additional comments are attached.

Reviewer #2: I have carefully reviewed the study titled "Fipronil In Sub-Lethal Doses Leads To Immuno Toxicological Effects In Broiler Birds". Although I find the scientific contribution of the article important, some deficiencies need to be addressed. I kindly request you to carefully consider the revisions in the attached file.

Reviewer #3: I have some trivial concerns regarding the manuscript. The points should be addressed in revision.

Line # 72-74. Repeated sentences.

Line # 103. Correct the word, “de-position”

Line # As the authors are going to evaluate sub-lethal dosage, therefore mention the LD50 for broilers with reference, to experimental birds and treatments sub-heading.

Line # 138-139. “Groups B, C, D, and E received fipronil @ 1.5, 2.5, 3.5, and 4.5 mg/kg body weight daily and these doses were selected on the basis of previous studies reported.” Add the reference.

Line # 154-155. What is meant by post-treatment, please re-write the sentence and make it clear.

Line # 182 …… chicks/group received ink (1 mL/kg) injection…. Please mention the type of ink.

Line # 185-186. How much amount of blood samples were mixed in 4mL of 1% sodium citrate?

Line # 238. ……. response (4.00 ± 1.00, however, IgG in birds of Groups……….

Close the parentheses properly.

Line # 371-375. Re-write the sentence.

Line # 377 and many others. Write SRBCs instead of SRBC.

Line # 393. As only twice the word DTH has been used in the whole MS, therefore use its full form.

Line # 397-399. In these sentences, mention the treatment groups.

Line # 409. The result revealed…….. Correct as “the results reviled ………”.

Line # 412 and 415. Add suffix th with numbers.

Line # 438. Re-write the sentence.

Line # 444-446. Improve the sentence.

Line # Write the full name of O. carvalhoi

6. PLOS authors have the option to publish the peer review history of their article (what does this mean?). If published, this will include your full peer review and any attached files.

Reviewer #1: No

Reviewer #2: No

Reviewer #3: **Yes: **Hafiz Iftikhar Hussain

---

## [Author Response · Author response to Decision Letter 0]

2 Dec 2024

We have provided File "Check List for Revision PONE-D-24-46735" stated how we dealt all points raised.

---

## [Editor Report · Decision Letter 1]

4 Dec 2024

FIPRONIL IN SUB-LETHAL DOSES LEADS TO IMMUNO-TOXICOLOGICAL EFFECTS IN BROILER BIRDS

PONE-D-24-46735R1

Dear Dr. Khan,

We’re pleased to inform you that your manuscript has been judged scientifically suitable for publication and will be formally accepted for publication once it meets all outstanding technical requirements.

Kind regards,

Adeel Sattar, Ph.D

Academic Editor

PLOS ONE
---

## [Editor Report · Acceptance letter]

17 Jan 2025

PONE-D-24-46735R1 

PLOS ONE

Dear Dr. Khan, 

I'm pleased to inform you that your manuscript has been deemed suitable for publication in PLOS ONE. Congratulations! Your manuscript is now being handed over to our production team.

Kind regards, 

on behalf of

Dr. Adeel Sattar 

Academic Editor

PLOS ONE